# Data mining and spatio-temporal characteristics of urban road traffic emissions: A case study in Shijiazhuang, China

**Lili Ren**[1], **Xuliang Guo**[2], **Jiangling Wu**[1]*, **Amit Kumar Singh**[3]

**1** School of Civil Engineering and Architecture, Henan University, Kaifeng, China, **2** School of Transportation, Jilin University, Changchun, China, **3** Consor Engineers, Austin, Texas, United States of America

* wujiangling_2006@126.com

**Data Availability Statement:** All relevant data are within the paper and its Supporting Information files. The data used in this study include the OSM (OpenStreetMap) road network vector data obtained from https://www.openstreetmap.org.

## Abstract

Accurate estimation of traffic emissions and analysis of spatio-temporal distribution on urban roads play a crucial role in the development of low-carbon transportation system. Traditionally, a region's emission characteristics have been studied using numerous emission models with GPS-based spatio-temporal data. Due to the heavy data processing needs of GPS-based data, emission characteristics for a large region have been studied by dividing the region into a limited number of smaller areas or units. Additionally, GPS data are based on a few vehicles in the traffic which does not fully reflect road conditions. This paper proposed an approach that can be used to study and calculate the spatio-temporal emission pattern of a region at a roadway section level by using Baidu's online traffic data and COPERT model. The proposed method can be used to estimate road-level emission patterns while avoiding the impact of redundant data in large datasets, making the dataset more reliable, applicable, and scalable. The proposed approach has been demonstrated through a study of spatio-temporal emission patterns in the Qiaoxi district within city of Shijiazhuang, China. Online data crawling technology was used to obtain data on urban road traffic speed and driving distance. The linear reference technology was used to construct a two-layer road network model to conduct the coupling and matching of traffic data with the road network data. The COPERT model was implemented to calculate the average traffic emissions on each road in the road network, and a traffic emission intensity index was proposed to quantify the CO, VOC, $NO_x$ and $CO_2$ emissions on urban roads in the study area. The analysis results show that the traffic emission intensity of the expressway, trunk road, secondary road, and branch road is high during the morning peak (7 AM-9 AM) and evening peak (5 PM—7 PM). The sections with higher traffic emission intensity are mainly concentrated on the main roads and secondary roads such as Jiefang South Street, Shitong Road and Xinhua Road. Nearly one-third of 2nd Ring and 3rd Ring roads also have relatively high emission intensity. The research results provide new ideas for estimating traffic emissions in urban road networks and analyzing the spatio-temporal distribution of traffic emissions. The research results can also provide a decision-making basis for traffic management departments to formulate energy-saving and emission-reduction measures and promote the development of urban green and low-carbon transportation.

The Road traffic data used in the research results is within the Supporting Information (S4_File).

**Funding:** The author(s) received no specific funding for this work.

**Competing interests:** The authors have declared that no competing interests exist.

## Introduction

In recent decades, greenhouse gas and air pollutant emissions have attracted much attention due to their adverse effects on climate change and human health, and transportation is an industry whose emissions of greenhouse gases and air pollutants are growing rapidly. With the economy's rapid development and the urban population increase, motor vehicle ownership in China has reached 417 million as of 2022 [1]. The increase in the number of motor vehicles has caused a sharp increase in the emissions of greenhouse gases and air pollutants, which has further intensified air pollution in Chinese cities [2, 3]. Air pollution adversely affects human health [4–6], causing millions of premature deaths yearly [7, 8].

Recently, the issue of urban traffic emissions and air pollution has become the focus of the public, and many institutions and researchers focus on traffic emission estimation and spatio-temporal distribution. In terms of traffic emissions estimation, researchers have developed several methods and models to analyze vehicle emissions and assess the impact of traffic on air pollution. Representative microscopic emission models include Comprehensive Modal Emission Model (CMEM), International Vehicle Emissions (IVE), and Motor Vehicle Emission Simulator (MOVES). CMEM was developed by the University of California using experimental vehicle exhaust emissions data [9]. This model estimates exhaust emissions from the vehicle's second-by-second speed trajectory data. IVE [10, 11] uses Vehicle Specific Power (VSP) and Engine Stress (ES) as input to calculate emission factor. MOVES [12] was developed by the U.S. Environmental Protection Agency (EPA) and used to estimate vehicle emissions (including greenhouse gases and toxic pollutants). Microscopic emission models can measure vehicle emissions more accurately. However, a large amount of measurement data is required in the research process. Therefore, the micro models are not suitable for traffic emission studies at the city level. Emission models suitable for macro and mesoscopic environments include Computer Program to calculate Emissions from Road Transport (COPERT), Emission Factor (EMFAC), and Mobile Source Emission factor (MOBILE). These models have the characteristics of fewer input parameters and convenient calculation. COPERT [13], a commonly used emission model in Europe, uses a large amount of experimental data to determine the emission parameters of road transport and obtain an emission inventory. EMFAC is used to calculate conventional exhaust and greenhouse gas emissions in the California region and was developed by the California Air Resources Board (CARB). This model is a macro-emissions model. MOBILE is a macro emission model launched by the EPA in 1978 [14]. The vehicle mileage, traffic volume, average speed, average mileage, and fuel type were used to calculate emissions [15–19]. Jamshidnejad et al. [20] proposed a general framework combining micro-emission and macro-emission models to estimate vehicle emissions from integrated traffic flows.

To study the spatio-temporal distribution of traffic emissions, researchers have combined vehicle travel data to analyze the temporal and spatial differences in urban traffic emissions. Luo et al. [21] used GPS data and GIS technology to analyze the spatio-temporal distribution of taxi energy consumption and emissions in Shanghai and found that the areas with high traffic emissions and fuel consumption were the downtown area and the area around Hongqiao. Nyhan et al. [22] calculated Singapore's traffic emissions based on GPS data and analyzed the spatio-temporal distribution of emissions using the grid as a unit, and finally determined that the core area of the urban area and the northern part of the central urban area were high-emission areas. Li et al. [23] used GPS data and the COPERT model to calculate the emissions of online car-hailing in Chengdu and analyzed the spatio-temporal distribution of online car-hailing emissions by dividing grid cells, the results show that the areas with higher emissions are located in the city's Second Ring Road, Shudu Avenue, and some intersections. Based on Didi data and survey data, Sun et al. [24] used the COPERT model to calculate the NOx

emissions of road sections, and obtained the distribution of road NOx emissions within 24 hours. Zhao et al. [25] studied the spatio-temporal distribution of $CO_2$ emissions during peak hours for different purposes and found that areas with high carbon levels are mainly located on a few arterial roads.

Many emission models and approaches have been developed to estimate traffic emissions and spatiotemporal distribution, but few studies have explored urban traffic emissions from the perspective of road segments. Although Meng et al. [26] considered road section variations for urban traffic emissions, the research data used are vehicle trajectory data and spatio-temporal traffic data extracted from GPS data. However, due to privacy and security protection policies and other reasons, the traffic data set and vehicle trajectory data set cannot be obtained difficulty, the spatiotemporal traffic data and vehicle trajectory data collected exhibit missing or corrupted data. Researchers have performed interpolation work on missing and damaged traffic data [27] and used deep learning models to regenerate vehicle trajectory data [28], but there are still deficiencies in fully reflecting road conditions, and the data processing workload is heavy. Crawling data based on online maps is a relatively convenient data collection method. For example, traffic data reflecting road conditions can be mined through Baidu online map. If large-scale and full-coverage traffic data are mined through the Baidu online map, and these traffic data are used for traffic emission research, the efficiency and accuracy of urban traffic emission estimation will be improved. Therefore, there is a gap in using the traffic data from Baidu online map and OSM roadway data to study urban traffic emissions and their spatio-temporal distribution characteristics.

To bridge this gap, this paper proposed an approach to study the spatio-temporal characteristics of urban traffic emissions from the road segment level using traffic data from the Baidu online traffic map and OSM roadway data. The main tasks of this study can be divided into three parts. First, the traffic data based on the Baidu online map and OSM road network data are used to build the double-layer data model of Shijiazhuang's road network. Then, combined with the COPERT model, a traffic emission intensity index is proposed to reflect the emission conditions of road sections. Finally, pollutant emission intensity and greenhouse emission intensity of urban roads are analyzed based on the time and space dimensions. The greenhouse considered is carbon dioxide ($CO_2$), and the pollutants considered are carbon monoxide (CO), atmospheric volatile organic compounds (VOC), and nitrogen oxides ($NO_x$).

Our main contributions can be summarized as follows:

1. We proposed to use crawled Baidu online map traffic data and OSM road network data to construct a two-layer road network data model for traffic emission research, which effectively avoids the impact of redundant data in big data and reduces the cost of data collection and processing.

2. This study provides new ideas for urban transportation emissions research and helps provide a decision-making basis for promoting urban green and low-carbon transportation development.

The remainder of this paper is organized as follows: Section 2 describes the data collection, data processing, and emission estimation methods for this study. Section 3 analyzes the results and discusses the findings. The final section summarizes the contributions and implications of this study.

## Methodology

The proposed methodology provides an approach for data collection and matching, emission estimation and generating spatio-temporal emission patterns.

### Data collection

The OpenStreetMap (OSM) road network vector data are obtained using the following process:

### (1) Road network vector data

The road network vector data is obtained through the OSM platform, which has high road positioning accuracy and topological relationship quality, and stores road network spatial data and attribute data. It is the most popular spontaneous geographic information [29]. Spatial data includes the longitude and latitude of the road's starting point and ending point; Attribute data includes road ID, road name, road type, starting point ID, ending point ID, road length, and driving direction. In order to utilize OSM data to use for emission calculations, the urban intersections data needs to be simplified to multiple line segments and nodes. Additionally, several road vector data that deviates from the actual road needs to be corrected. This provides convenience for road traffic data acquisition and road network topology processing.

### (2) Road traffic data

The road traffic data is mined through Baidu's online map API. Traffic data includes: road ID, sampling time, road length, travel time, and average speed for every 15 minutes interval. Occasionally, data loss within Baidu's system occurs due to network delay and this may affect traffic data quality. Linear interpolation method can be used to fill in the missing traffic data.

### Double-layer road network model

In this study, graph theory and line reference model are used to build a double-layer road network model, the structure of which is shown in (Fig 1). This model can match road network vector data with road traffic data, enhance data expression and analysis capabilities, and provide a data basis for traffic emission research.

The double-layer road network model consists of two parts, namely road network topology and road network event table, as follows:

### (1) Road network topology

Based on graph theory, the intersections and road segments of the network are topologically processed into corresponding nodes and links, and the topological relationship is represented by an adjacency matrix. (Fig 2) shows an example of a road network topology and its adjacency matrix. The properties of nodes and links are consistent with those of intersections and road segments, as shown in (S1 Table).

### (2) Road network event table

Based on the linear referencing technology, the road network event table is constructed, which is used to store traffic information. Linear referencing technology associates multiple attributes with linear features through an event table. The linear reference event of the road can be changed by changing the information of the event table, which is suitable for traffic data with spatio-temporal characteristics. Therefore, this paper uses traffic data as line events to generate event tables and uses linear referencing techniques to relate line events to roads in the topology. The line events information is shown in (S2 Table), and the line reference processing is shown in (Fig 3).

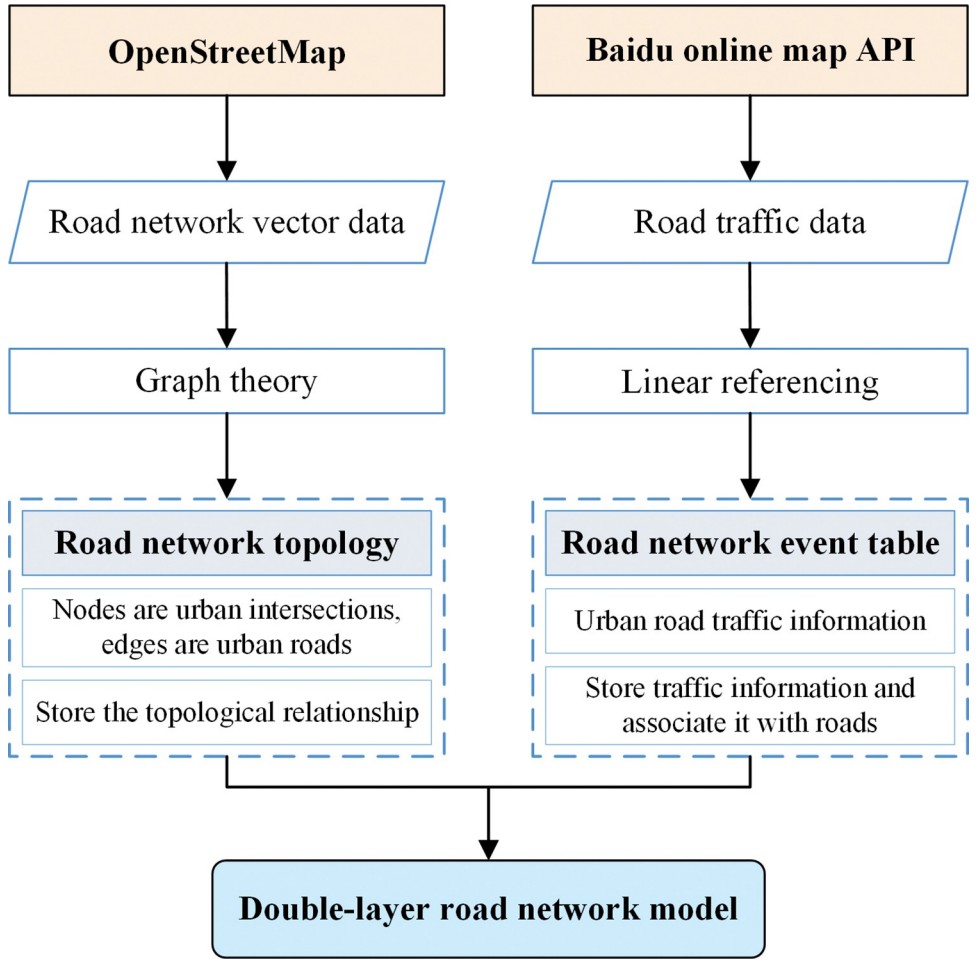

**Fig 1. The framework diagram of road network model.**

## Emission estimation

The COPERT model is a vehicle emission calculation model that supports the European emission standard, and speed is used as an input parameter to calculate vehicle emissions. China's emission standard is similar to Europe, and studies have demonstrated the applicability of the

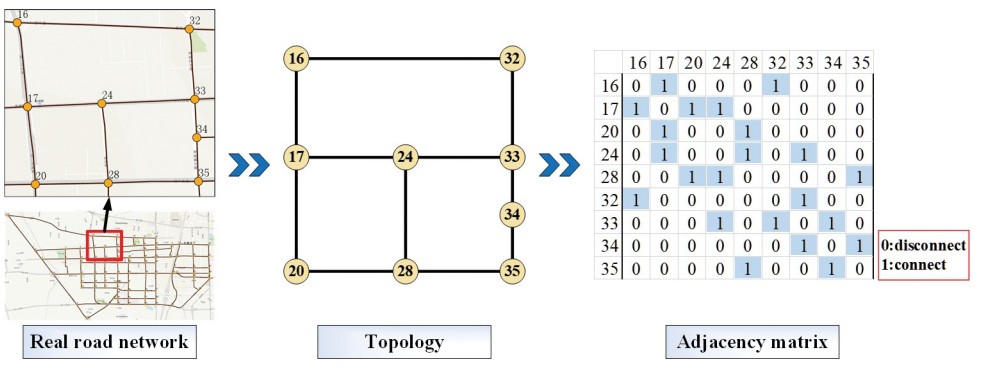

**Fig 2. Example of road network topology.**

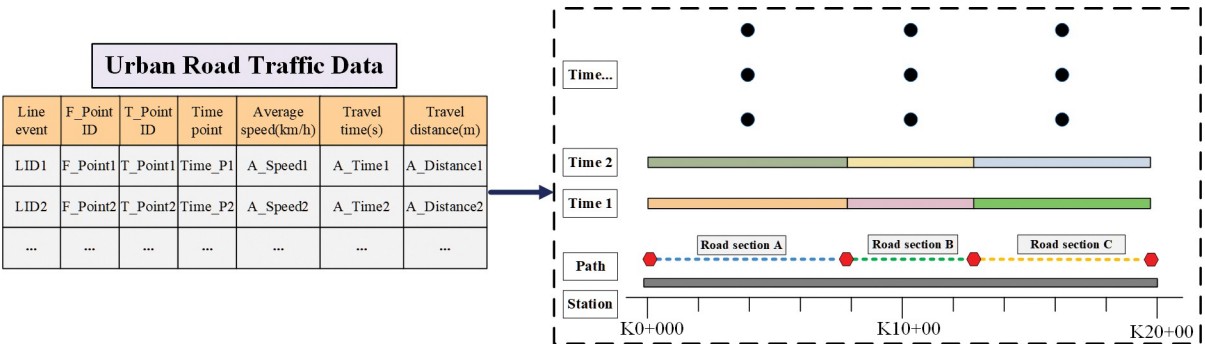

**Fig 3. Linear referencing processing of urban road traffic data.**

COPERT model in China [23, 30]. The latest version of this model is suitable for motor vehicles that meet the Euro VI emission standard [31], and the current emission standard, National Phase VI Motor Vehicle Pollutant Emission Standards in China, used in Shijiazhuang, are basically the same as the Euro VI emission standard. Therefore, this paper approximately considers that the vehicle emission factors for Shijiazhuang are the same as which are used in COPERT.

The cold and hot emissions of motor vehicles can be calculated based on the COPERT model. Due to the small proportion of cold emissions [23], this paper only considers the thermal emissions of vehicles. The heat emission is related to emission factor and driving distance [31], as shown in Eq (1).

$$Q_{P,j} = E_{P,j} \bullet L_j \tag{1}$$

where $Q_{P,j}$ is the vehicle pollutant emission on the $j_{th}$ road, $L_j$ is the driving distance of the vehicle on the $j_{th}$ road, and $E_{P,j}$ is the pollutant emission factor of the $j_{th}$ road (g/km). The relevant parameter for the emission factor is the average speed, as shown in Eq (2).

$$E_{P,j} = \frac{\alpha_P \cdot v_j^2 + \beta_P \cdot v_j + \gamma_P + \delta_p/v_j}{\varepsilon_P \cdot v_j^2 + \zeta_P \cdot v_j + \eta_P} \tag{2}$$

Where $v_j$ is the average speed of motor vehicles on the road j (km/h), $\alpha_P, \beta_P, \gamma_P, \delta_p, \varepsilon_P, \zeta_P, \eta_P$ are parameters related to vehicle type, fuel type, emission standard, and engine type.

Most vehicles in the urban area of Shijiazhuang are gasoline vehicles that meet the National VI emission standard. To simplify the calculation process, this paper sets the vehicle type as small cars, the fuel type as gasoline, and the emission standard as Euro VI (equivalent to China VI). The corresponding COPERT parameters are shown in (Table 1). Among them, the

**Table 1. Parameters for emission factors in the COPERT model [31].**

| Pollutant$_P$ | CO | VOC | NO$_X$ | CO$_2$ |
|---|---|---|---|---|
| $\alpha_P$ | 8.51E-4 | 3.814E-6 | -3.145E-4 | 3.325E-1 |
| $\beta_P$ | -1.799E-1 | -7.074E-4 | 1.031E-1 | -1.756E+1 |
| $\gamma_P$ | 1.13E+1 | 4.525E-2 | 2.391E-1 | 1.452E+3 |
| $\delta_p$ | 1.69E+1 | 1.731E-1 | -3.393E-1 | 1.761E-11 |
| $\varepsilon_P$ | 2.643E-3 | 6.99E-5 | 3.454E-2 | 8.009E-4 |
| $\zeta_P$ | -7.19E-1 | -4.754E-2 | 1.986 | 9.133E-2 |
| $\eta_P$ | 5.079 | 6.212 | 1.264 | 3.513 |

parameter of $CO_2$ emission factor is obtained by the parameter of energy consumption factor EC and conversion coefficient (69.3 g $CO_2$/MJ).

## Emission intensity index

Based on vehicle emissions and traffic volume, the total vehicle emissions and road traffic emissions can be estimated [17, 30]. However, it is very expensive to obtain traffic volume and classification information for all roads in cities. The average speed of vehicles can reflect the running status of road traffic, and the cost of mining traffic data through Baidu's online map API is low. Therefore, this paper combines the COPERT model to propose a traffic emission intensity index from the perspective of road sections, which can reflect the emission situation of urban roads and effectively reduce the cost of traffic data acquisition.

The meaning of traffic emission intensity is the traffic emission generated by vehicles traveling at the average speed and passing the road, that is, the traffic emission per vehicle, and the unit of traffic emission intensity is g/veh, as shown in Eq (3).

$$H_{P,j,t} = E_{P,j,t} \bullet L_j = \frac{\alpha_P \cdot v_{j,t}^2 + \beta_P \cdot v_{j,t} + \gamma_P + \delta_p/v_{j,t}}{\varepsilon_P \cdot v_{j,t}^2 + \zeta_P \cdot v_{j,t} + \eta_P} \bullet L_j \tag{3}$$

where $H_{P,j,t}$ is the traffic emission intensity on the $j_{th}$ road in the road network at the $t_{th}$ moment, $E_{P,j,t}$ is the motor vehicle emission factor of the $j_{th}$ road in the road network at the $t_{th}$ moment, $L_j$ is the distance traveled by the vehicle on the $j_{th}$ road, $v_{j,t}$ is the average speed of vehicles traveling on the $j_{th}$ road at the $t_{th}$ moment, and other parameters are the same as Eq (2).

Based on the road traffic data and the equation of traffic emission intensity, the hourly traffic emission intensity and the daily traffic emission intensity of each road can be calculated, as shown in Eq (4) and Eq (5).

$$H_{P,j}^h = \frac{1}{T_h} \sum_{t=1}^{T_h} H_{P,j,t} \tag{4}$$

$$H_{P,j}^d = \frac{1}{T_d} \sum_{t=1}^{T_d} H_{P,j,t} \tag{5}$$

where $H_{P,j}^h$ is the hourly traffic emission intensity, $H_{P,j}^d$ is the daily traffic emission intensity, $T_h$ is the data collection times per hour, and $T_d$ is the data collection times per day.

## Case study of Shijiazhuang City

The proposed methodology was used to study the spatio-temporal emission pattern for the Qiaoxi district in the southwest of Shijiazhuang city, China, shown in (Fig 4). Qiaoxi district is the political, economic, cultural, educational, technological, and financial center of Shijiazhuang City, with a total population of 970,000 and a total area of about 76 square kilometers. Qiaoxi district has a relatively developed transportation network and can be integrated into the Beijing-Tianjin-Hebei 1-hour "traffic circle-economic circle".

The data used in this study include the OSM (OpenStreetMap) road network vector data obtained from https://www.openstreetmap.org and the road traffic data mined from the Baidu online map API. In order to make the OSM data more consistent with the research, the urban intersections data was simplified to have multiple line segments and nodes. Additionally, the road vector data that deviates from the actual road were corrected. The road traffic data used in this study was mined through Baidu's online map API. The data involved 153 urban roads

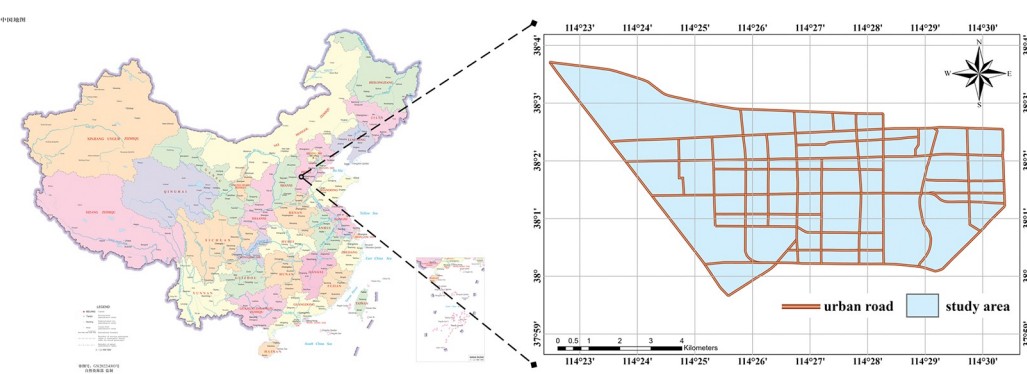

**Fig 4. The study area.**

and 102,816 data entries over seven days (October 10 to 17, 2022), and the data interval was 15 minutes. Traffic data includes collection time, driving distance, travel time, and average speed. Traffic demand for the weekdays was similar. Due to the relatively stable travel demand on weekdays, Wednesday, October 12, 2022, a typical weekday, was selected to analyze the spatio-temporal distribution characteristics of urban road traffic emissions within 24 hours. To avoid data loss caused by network delay and affect data quality, this paper uses the linear interpolation method to fill in the missing data.

## Spatial distribution analysis of traffic emission intensity

Based on the traffic speed data and road length data in the double-layer road network model, the traffic emission intensity of each road in the study area at different times is calculated through Eq (2) and Eq (3). Then the daily traffic emission intensity of CO, VOC, NOx, and $CO_2$ for the 153 roads in the study area is calculated according to Eq (5). Use SPSS data analysis software to conduct descriptive statistics on all-day traffic emission intensity of all roads, as shown in (Table 2).

(Table 2) shows that the maximum traffic emission intensity of CO, VOC, $NO_x$, and $CO_2$ is 1.7817g/veh, 0.0486g/veh, 0.2201g/veh, 1201.042g/veh, and the minimum is 0.034g/veh, 0.0012g/veh, 0.0059g/veh, 33.3489g/veh, the average traffic emission intensity of CO, VOC, $NO_x$, $CO_2$ of all roads throughout the day is 0.2136g/veh, 0.0065g/veh, 0.0311g/veh, 171.8589g/veh.

**Table 2. Descriptive statistics of urban road traffic emission intensity.**

| Statistical indicators | Traffic Emission Intensity (g/veh) | | | |
|---|---|---|---|---|
| | CO | VOC | $NO_x$ | $CO_2$ |
| Max | 1.7817 | 0.0486 | 0.2201 | 1201.042 |
| Min | 0.034 | 0.0012 | 0.0059 | 33.3489 |
| Mean | 0.2136 | 0.0065 | 0.0311 | 171.8589 |
| Rang | 1.7477 | 0.0474 | 0.2142 | 1167.6931 |
| Standard Deviation | 0.2221 | 0.0063 | 0.0299 | 161.9396 |
| Variance | 0.049 | 0.0001 | 0.0009 | 26224.4507 |
| First Quartile | 0.1248 | 0.0038 | 0.0176 | 97.6441 |
| Second Quartile | 0.1681 | 0.0052 | 0.0251 | 137.4231 |
| Third Quartile | 0.2076 | 0.0064 | 0.0311 | 173.7857 |

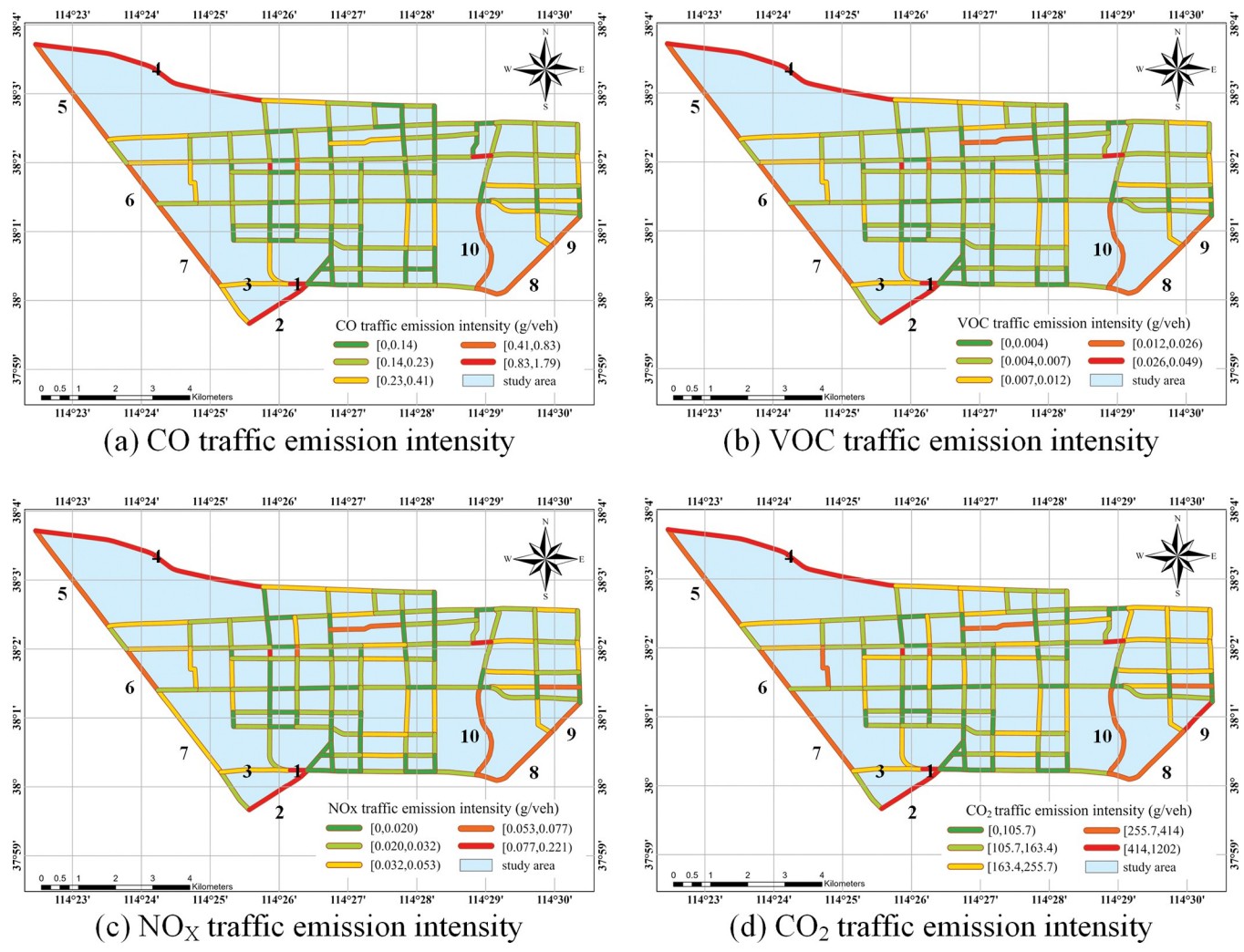

**Fig 5. Spatial distribution of traffic emission intensity on urban road network.**

Based on the full-day traffic emission intensity of 153 roads, the natural discontinuity method is used for grouping and summarizing. Combined with GIS technology, the emission intensity of roads is visualized spatially, as shown in (Fig 5). The emission intensity grouping is represented by different colors.

According to the (Fig 5), it can be seen that the traffic emission intensity of CO, VOC, $NO_x$, and $CO_2$ of urban roads in the study area present a similar spatial distribution. Roads with high emission intensity are mainly West 2nd Ring South Road, Shitong Road, Xinhua Road, Yuquan Road, South 2nd Ring West Road, Yuhua Road, Huitong Road, Jiefang South Street, West 3rd Ring Road, Chengjiao Street, Changqing Road, etc. The sections with prominent traffic emission intensity are shown in (S3 Table).

According to the field survey, the West 3rd Ring Road, West 2nd Ring South Road, and South 2nd Ring West Road are all expressways in Shijiazhuang City, which are important traffic circles. Jiefang South Street is the main road, and Shitong Road, Xinhua Road, and Huitong Road are secondary roads. The specific analysis are as follows:

1. Yuquan Road (the intersection of West 3rd Ring Road and Yuquan Road to the intersection of West 2nd Ring and South 2nd Ring Road) intersects with the urban ring road, has many

ramps, and the traffic facility structure is relatively complex. Many vehicles from West 2nd Ring Road South, South 2nd Ring West Road, and West 3rd Ring Road will drive on Yuquan Road. However, the higher speed of these vehicles can easily cause traffic congestion on Yuquan Road, which in turn leads to higher traffic emission intensity.

2. South 2nd Ring West Road (the intersection of West 2nd Ring Road and South 2nd Ring Road to the intersection of Shitong Road and South 2nd Ring Road) intersects with West 2nd Ring South Road, Yuquan Road and Shitong Road in a three-dimensional way. The road length is relatively short and there are many interchange ramps. So the congestion is serious and the emission intensity is high.

3. The West 3rd Ring Road is an urban expressway with a relatively high average speed, but there are only ramp connections between Huaian Road and Yuhua Road and other main roads. The amount of emissions is generated when passing-by is large, so the traffic emission intensity of the West 3rd Ring Road is relatively high.

4. There are a large number of residential areas, schools, and transportation hubs along Xinhua Road, such as Shijiazhuang Xiwang Passenger Station, Hebei International Preparatory College, No. 41 Middle School, etc. The large volume of motor vehicle trips in these places leads to a large traffic volume and low average speed on Xinhua Road, which in turn increases the intensity of traffic emissions.

5. Both South 2nd Ring Road and West 3rd Ring Road have overpasses with Shitong Road, and many residential areas along Shitong Road. Complex traffic flow structures, traffic channelization, and overpass ramps can all aggravate road traffic congestion and lead to high traffic emission intensity.

6. There are many signalized intersections along the Huitong Road, and there are many residential areas, parks, commercial hotels, etc. on both sides of the road, and the road queuing time is long in peak hours. Jiefang South Street runs through the entire municipal area, especially Jiefang South Street (the intersection of Jiefang South Street and the South 2nd Ring Road to the intersection of Jiefang South Street and Huaian Road) passes through Shijiazhuang Railway Station, and the traffic flow is huge and complex. Therefore, the average speed on Huitong Road and Jiefang South Street are low and emission intensity on these roads are high.

### Temporal distribution analysis of traffic emission intensity

Based on the traffic speed data and road length data in the double-layer road network model, the hourly traffic emission intensity of the road is obtained using Eq (2), Eq (3), and Eq (4), and the change trend and distribution characteristics of emission intensity within 24 hours are analyzed. Since the variation in traffic patterns of different roadway types during the day differs from each other, this paper analyzes the time distribution characteristics of traffic emission intensity for different types of roads (urban expressway, urban trunk road, urban secondary road, and urban branch road). The summary of the emission pattern is provided in (Table 3).

The detailed emission pattern results by roadway type are as follows:

(1) Temporal distribution of traffic emission intensity on urban expressway

(Fig 6) shows that the emission intensity of expressway traffic is in a flat state from 0:00 to 6:00, and only peaks at 9AM to 10 AM and 5 PM to 6 PM, but the overall increase is not large. The reason may be that only a small number of people need to travel through the expressway during the morning and evening peak hours because of the long distance between the place of

**Table 3. Changes in traffic emission intensity of different types of urban roads.**

| State | Urban Expressway | Urban Trunk Road | Urban Secondary Road | Urban Branch Road |
|---|---|---|---|---|
| Steady | 0 AM-6 AM; 11 AM-4 PM; 10 PM-12 PM | 0 AM-5 AM; 10 PM-12 PM | 0 AM-7 AM; 8 PM-12 PM | 0 AM-6 AM; 10 AM-1 PM; 11 PM-12 PM |
| Rising | 6 AM-8 AM; | 5 AM-8 AM; | 7 AM-9 AM; | 6 AM-8 AM; |
|  | 4 PM-5 PM | 1 PM-5 PM | 1 PM-5 PM | 1 PM-6 PM |
| Peak | 9 AM-10 AM; | 8 AM-9 AM; | 9 AM-10 AM; | 8 AM-9 AM; |
|  | 5 PM-6 PM | 5 PM-6 PM | 5 PM-6 PM | 6 PM-7 PM |
| Falling | 10 AM-11 AM; | 9 AM-1 PM; | 10 AM-13 AM; | 9 AM-10 AM; |
|  | 6 PM-9 PM | 6 PM-10 PM | 6 PM-8 PM | 7 PM-11 PM |

work and the place of residence. At noon, most travelers do not pass through the expressway when they travel for dining or entertainment, and the average speed of the expressway changes little; The expressway has a high capacity and a large number of lanes, and the increase of some vehicles will not have a great impact on the average speed of the road, so the emission intensity changes little.

(2) Temporal distribution of traffic emission intensity on urban trunk road

(Fig 7) shows that the emission intensity of trunk road is in a stable state from 0 AM to 5 AM, which may be few vehicles on the road and the vehicles keep a high speed. The emission intensity increases from 5 AM, and reach the peak in the morning from 8 AM to 9 AM, which may be due to the gradual increase in traffic counts and the average speed decrease during the morning peak. The emission intensity gradually decreased from 9 AM to 1 PM, which may be due to the decrease in travel volume and the increase in the average road speed during this period during working hours and dining hours. The emission intensity increases again from 1

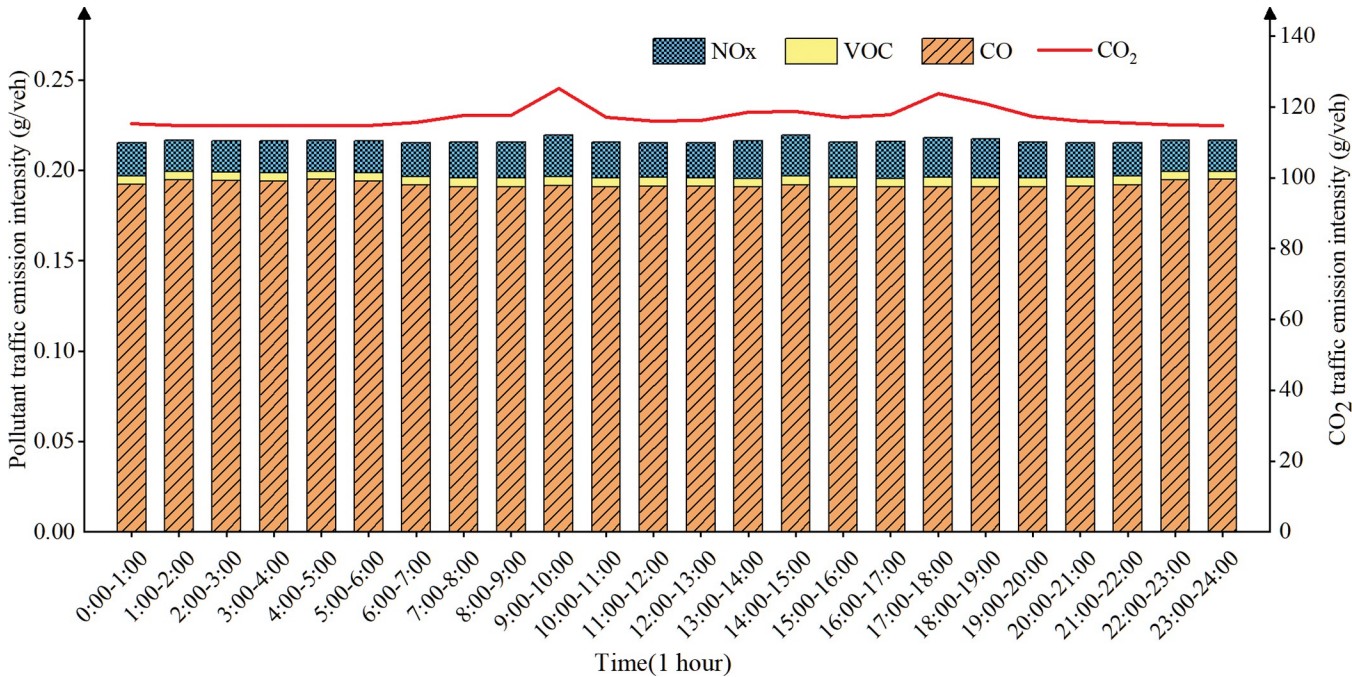

**Fig 6. Temporal distribution of urban expressway traffic emissions.**

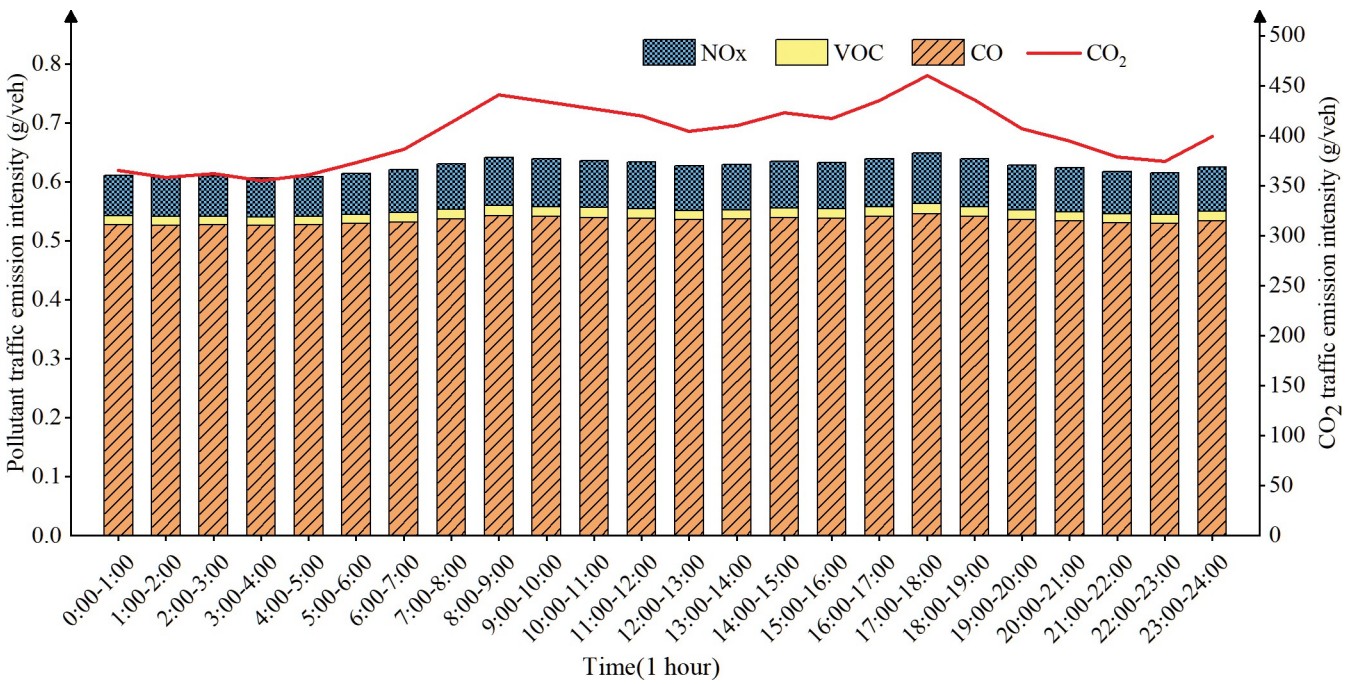

**Fig 7. Temporal distribution of urban trunk road traffic emissions.**

PM and reach the peak in the afternoon from 5 PM to 6 PM. It may be more passenger and freight vehicles in the afternoon, congestion is severe and average speed is low during the evening peak period, which aggravates traffic emissions. From 6PM to 8 PM, the emission intensity decreases but remain at a relatively high level, which may be the lower average road speed caused by nighttime recreational activities. From 8 PM to 12 PM, most of the driving residents have gone home, there are fewer vehicles on the road, which leads to an increase in road speed and a reduction in traffic emission intensity.

(3) Temporal distribution of traffic emission intensity on urban secondary road

(Fig 8) shows that the emission intensity of secondary roads is at a relatively high level. The emission intensity increased significantly from 7 AM and reached the peak in the morning from 9 AM to 10 AM, which may be the gradual decrease of the average speed and the gradual increase of vehicles during the morning peak period. The emission intensity decreased significantly from 10 AM to 1 PM, which may be due to the time for working and dining, the travel volume decreased and the average speed increased. The emission intensity increased from 1 PM and peaked again at 5 PM-6 PM The reason may be that there are more passenger and freight transport vehicles in the afternoon, and congestion is severe in the evening commute rush hour, which leads to average speed drops to the lowest value. From 18:00 to 24:00, as most people go home, the number of vehicles on the road decreases, and the average speed gradually increases, resulting in a gradual decrease in emission intensity.

(4) Temporal distribution of traffic emission intensity on urban branch road

(Fig 9) shows that the emissions intensity increases and decreases significantly from 6 AM to 12AM, and reaches a peak from 8AM to 9 AM. It may be that the number of motor vehicles and non-motor vehicles on the branch road during the morning rush hour increases, and the average speed decreases. As the morning traffic rush hour ends and there are fewer vehicles on

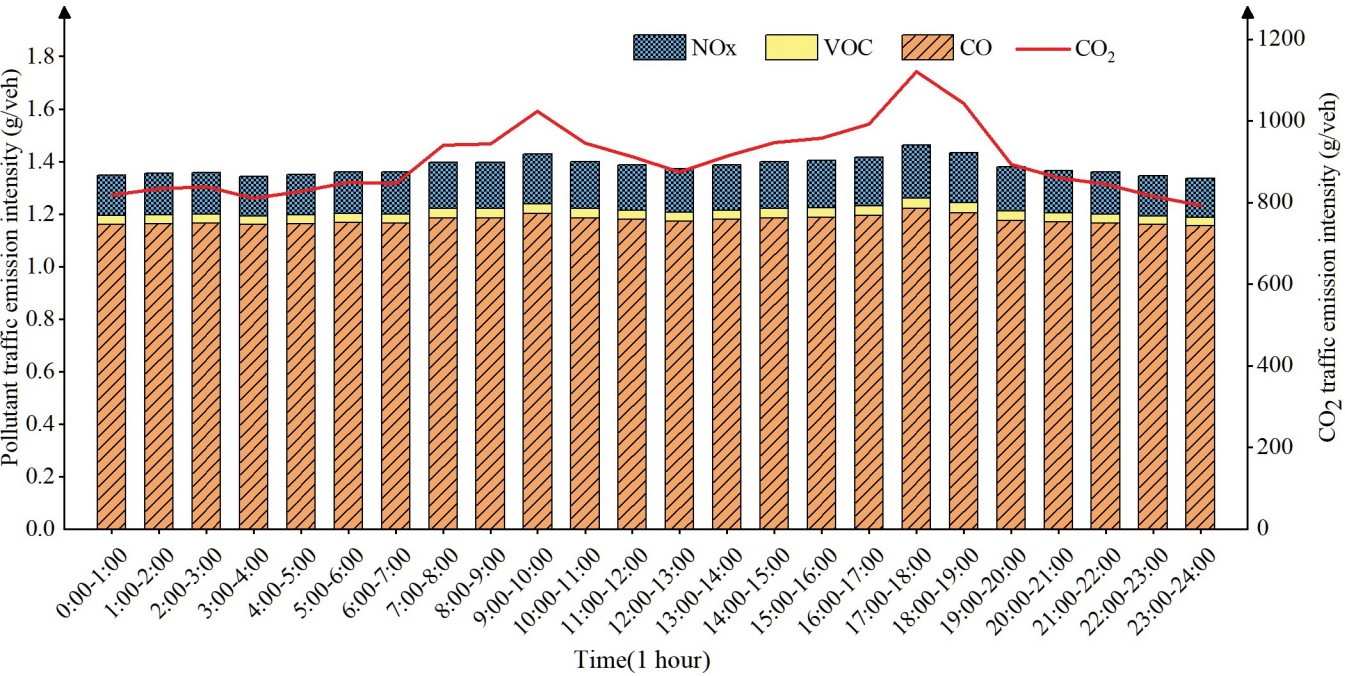

**Fig 8. Temporal distribution of urban secondary road traffic emissions.**

the road, the average speed increases again. The emission intensity gradually increases to a higher level again from 12AM, and reach the afternoon peak at 6 PM -7 PM, which may be due to the gradual increase in the number of non-motor vehicles on branch road and the

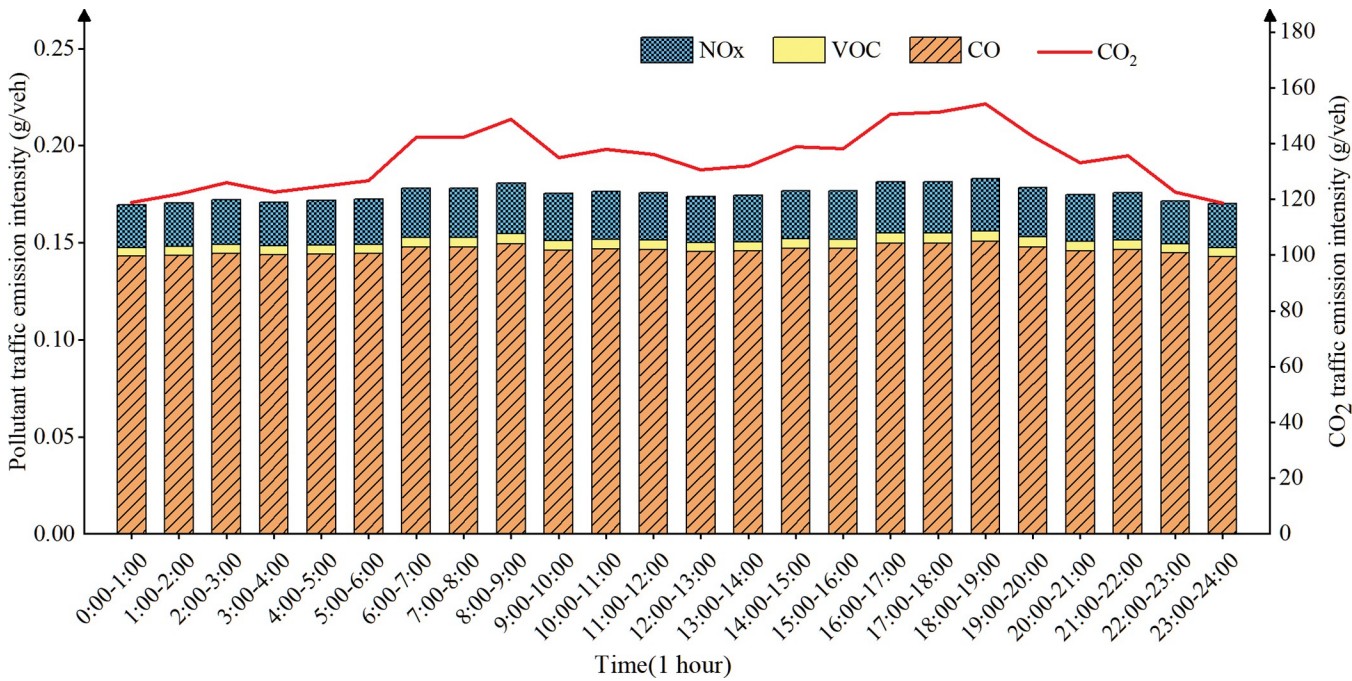

**Fig 9. Temporal distribution of urban branch road traffic emissions.**

mixed traffic is serious, so the vehicle' speed decreases and reaches the valley value branch during the evening peak. From 7 PM to 12 PM, non-motor vehicles on branch roads decrease, and motor vehicles can drive at high speed on branch roads, thereby reducing the emission intensity gradually. To sum up, the traffic emission intensity of the four types of urban roads are different in different time periods, but the overall trend of emission intensity changes over time is consistent, and the emission intensity is evenly distributed between 7 AM-9 PM. Moreover, the emission intensity has obvious characteristics of peaks in the morning and evening, and the time to reach the peak roughly coincides with the time of the traffic peak in the morning and evening.

## Relationship between traffic emission intensity and speed

In order to explore factors that affect traffic emission intensity, this paper uses the above data to analyze the relationship between emission intensity and average speed. To avoid the impact of the difference in driving distance on the analysis, the theoretical and actual values of traffic emission intensity of CO, VOC, NOx, and $CO_2$ in per kilometer, on each road were calculated respectively, and their relationship curves were drawn, as shown in (Fig 10). Among them, X represents the average speed, Y represents the traffic emission intensity in per kilometer and $R^2$ is the goodness of fit.

As shown in (Fig 10) when the average speed is in the range of 10-80km/h, the CO, VOC, NOx, and $CO_2$ emission intensities on the roads in the study road network are: 0.220~0.253g/veh, 0.005~0.01g/veh, 0.018 ~0.047g/veh, 133.78~275.21g/veh, the relationship between them and the average speed can be well fitted by polynomial curve. Among them, the traffic emission intensity of CO decreases first and then increases with the increase of the average speed; the traffic emission intensity of VOC, $NO_x$ and $CO_2$ have a similar trend and decrease with the average speed increase. The difference in emissions at different speeds is related to the speed. When the average speed is low, that indicates the acceleration, deceleration and idling of vehicles on the road are more frequent at this time, which in turn causes a sharp increase in emissions [26]. And in the case of traffic congestion, the idle time of the vehicle exceeds 50%, which will cause a large amount of $CO_2$ emissions. Therefore, the average speed will affect the magnitude of traffic emission intensity, which in turn may affect the spatio-temporal distribution of traffic emission intensity.

## Conclusions

This paper proposes a method for estimating traffic emissions using big data, using the proposed traffic emission intensity index to quantify traffic emissions from the road level and study regional spatiotemporal distribution patterns. Based on the traffic data of the Baidu online map and OSM road network data, this paper used the COPERT model to estimate the CO, VOC, NOx, and CO2 emission intensity of urban roads in Qiaoxi district, Shijiazhuang City, and analyzed their temporal and spatial distribution characteristics. The conclusions are as follows:

1. From the perspective of temporal distribution, the traffic emission intensity on different types of urban roads has obvious morning and evening peak characteristics. The morning peak of emission intensity is 8AM-10 AM, 1–2 hours later than the traditional morning traffic peak. The evening peak period of emission intensity is 5 PM-7 PM, which is more consistent with the evening commuting peak.

2. From the perspective of spatial distribution, the road sections with high traffic emission intensity in the study road network are some road sections of the west 3rd ring road and

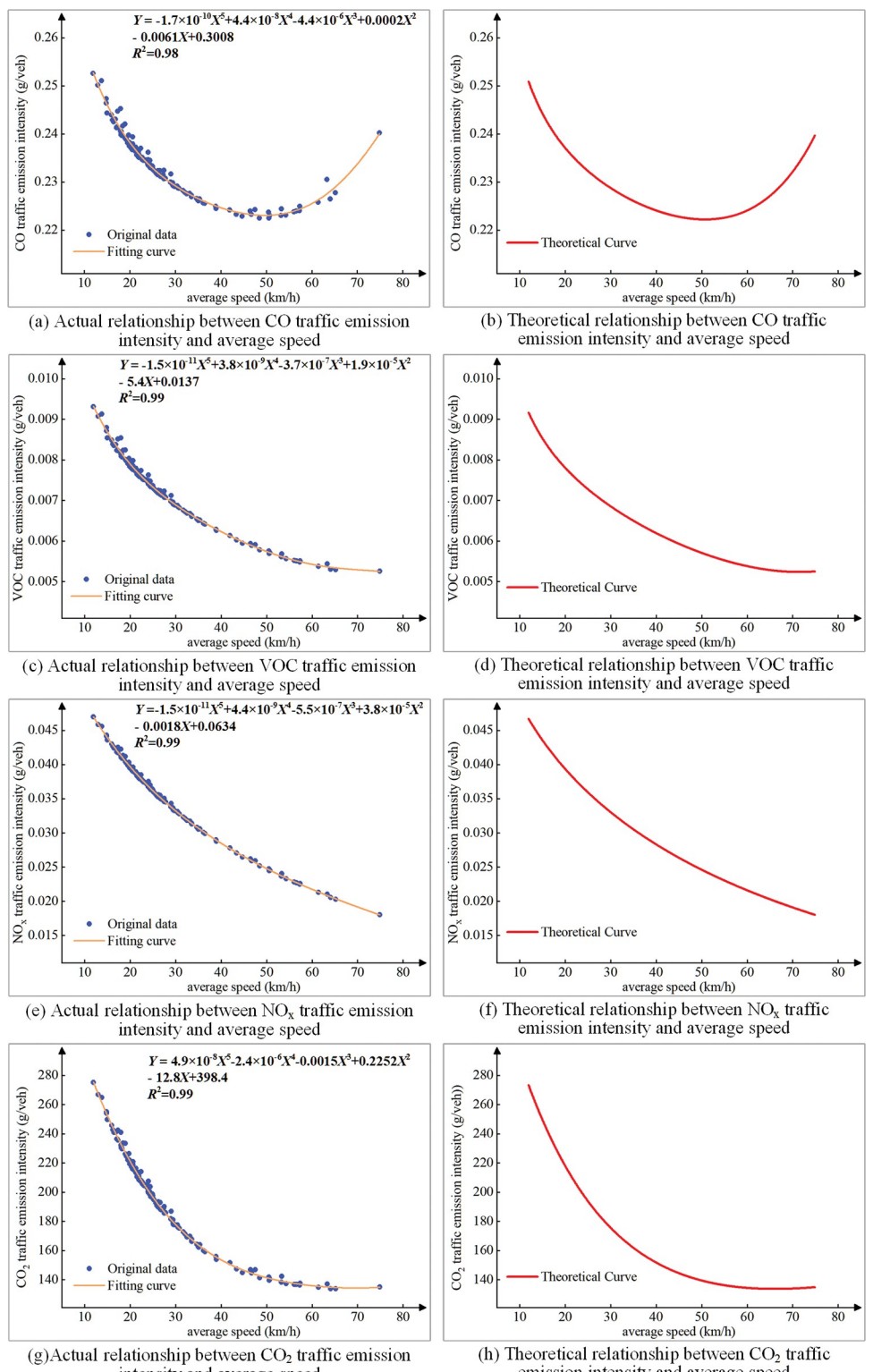

**Fig 10. The relationship between traffic emission intensity and average speed.**

west second ring road, the urban trunk road sections, and urban secondary road sections that intersect with ring roads (including Jiefang South Street, Xinhua Road, Shitong Road). The reason is that these roads sections have high traffic volume, low average vehicle speed, and high emissions.

3. With the increase of the average speed, the emission intensity per unit distance of VOC, NOX, and CO2 gradually decreases, the emission intensity per unit distance of CO decreases first and then increases, presenting a U-shaped distribution. Therefore, the traffic emission intensity of urban roads can be reduced through traffic induction optimization and speed limit measures.

This paper proposes that traffic emission intensity quantifies the emission of urban roads, reveals the temporal and spatial distribution characteristics and influencing factors of emission intensity, and provides a decision-making basis for traffic control departments to evaluate urban traffic emissions and formulate emission reduction measures. Due to data limitations, this paper only analyzes the emission intensity of the Qiaoxi district of Shijiazhuang in China, the result only reflects the urban road traffic emissions in this special area. Future research should expand the scale space of the data, and then analyze the temporal and spatial distribution characteristics of more representative urban road traffic emission intensity.

## Supporting information

**S1 File. Crawled road traffic data set from the Baidu online map.**
(XLSX)

**S1 Table. Attribute information of nodes and edges in topology.**
(DOCX)

**S2 Table. Line event information in linear reference models.**
(DOCX)

**S3 Table. Roads with high traffic emission intensity.**
(DOCX)

## Acknowledgments

The authors would like to express their sincere thanks to the anonymous reviewers for their helpful comments and valuable suggestions on this paper.

## Author Contributions

**Conceptualization:** Lili Ren, Xuliang Guo, Jiangling Wu, Amit Kumar Singh.

**Data curation:** Lili Ren, Xuliang Guo.

**Formal analysis:** Lili Ren, Xuliang Guo, Jiangling Wu.

**Methodology:** Lili Ren, Xuliang Guo, Jiangling Wu.

**Validation:** Lili Ren, Xuliang Guo.

**Visualization:** Lili Ren, Xuliang Guo, Jiangling Wu.

**Writing – original draft:** Lili Ren, Xuliang Guo, Amit Kumar Singh.

**Writing – review & editing:** Lili Ren, Xuliang Guo, Jiangling Wu, Amit Kumar Singh.

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
