## [Decision Letter · Decision Letter 0]

30 Aug 2023

PONE-D-23-14153Data Mining and Spatio-temporal Characteristics of Urban Road Traffic Emissions: A Case Study in Shijiazhuang, ChinaPLOS ONE

Dear Dr. Wu,

Thank you for submitting your manuscript to PLOS ONE. After careful consideration, we feel that it has merit but does not fully meet PLOS ONE’s publication criteria as it currently stands. Therefore, we invite you to submit a revised version of the manuscript that addresses the points raised during the review process.

We look forward to receiving your revised manuscript.

Kind regards,

Xiangjie Kong

Academic Editor

PLOS ONE

Journal Requirements:

3. We note that Figures 4 and 5 in your submission contain map images which may be copyrighted. All PLOS content is published under the Creative Commons Attribution License (CC BY 4.0), which means that the manuscript, images, and Supporting Information files will be freely available online, and any third party is permitted to access, download, copy, distribute, and use these materials in any way, even commercially, with proper attribution. For these reasons, we cannot publish previously copyrighted maps or satellite images created using proprietary data, such as Google software (Google Maps, Street View, and Earth). For more information, see our copyright guidelines: http://journals.plos.org/plosone/s/licenses-and-copyright.

    1. You may seek permission from the original copyright holder of Figures 4 and 5 to publish the content specifically under the CC BY 4.0 license. 

Reviewers' comments:

Reviewer's Responses to Questions

**Comments to the Author**

1. Is the manuscript technically sound, and do the data support the conclusions?

Reviewer #1: Yes

Reviewer #2: Yes

Reviewer #3: Yes

2. Has the statistical analysis been performed appropriately and rigorously? 

Reviewer #1: Yes

Reviewer #2: Yes

Reviewer #3: Yes

3. Have the authors made all data underlying the findings in their manuscript fully available?

Reviewer #1: Yes

Reviewer #2: Yes

Reviewer #3: Yes

4. Is the manuscript presented in an intelligible fashion and written in standard English?

Reviewer #1: Yes

Reviewer #2: Yes

Reviewer #3: Yes

5. Review Comments to the Author

Reviewer #1: 1. The map of China in Figure 4 is incomplete, and the South China Sea Islands are omitted.

2. What does “EF” in the second paragraph of the introduction mean?

3. The organization of the paper is somewhat chaotic. The last second paragraph of the introduction points out that Section 2 describes the data and data processing. But the following headline is "Methodology".

4. The authors should go through the paper for typos and syntax errors. There are many grammatical errors and expressing inconsistencies in the paper. For example: The symbols “-” and “~” in the sentence “0.220-0.253g/veh, 0.005-0.01g/veh, 0.018324 ~0.047g/veh, 133.78~275.21g/veh” are inconsistent.

5. What are the innovations of the paper? What are the contributions to this research field?

6. The first paragraph of the introduction is too simple, it is recommended to rewrite it.

Reviewer #2: This paper introduces a methodology for using big data sources to estimate emission intensity on roadways. The proposed methodology can replace the expensive methods of evaluating roadway emissions. Based on my review of this paper, here are my comments:

(1) Line 181 In equation 3, why is emission intensity directly related to length? What does this relationship mean? What is the unit of "Hp g/veh-km"? If the average speed is the same, would a road with a 2 km link have double the emission intensity of a road with a 1 km link?

(2) Line 87: Since traditional GPS-based emission analysis requires heavy data processing and specialized training, how easy is it to use the proposed method? If decision-makers intend to use the proposed methodology, how many days would it take to complete the analysis?

(3) Line 38: In the abstract, mention the actual number or percentage of segments on Second and Third Street instead of "some segments."

(4) Line 171: Please cite the original data source for Table 1.

(5) Line 208: What percentage of the traffic data was original, and what percentage of data was missing that required linear interpolation?

(6) Line 211: Provide further details on how Equation 4 was used in the case study. Was the average speed for every 15 minutes used?

(7) Figures 6-9: Why is CO emission almost constant regardless of the time of day? Is CO emission not affected by slowdowns during peak hours?

Reviewer #3: This paper identifies a gap in the existing literature related to using big data regarding traffic to evaluate traffic emissions. It aims to fill this gap by utilizing Baidu traffic data and the COPERT model to estimate traffic emissions.

(1) It's not clear what "emission intensity" means. Are we trying to identify roadways where emissions from a single vehicle are high? This reasoning seems to support not using traffic volume as a factor.

(2) Is emission intensity (emissions from a single car) high for congested roadways? Why does a vehicle stuck in congestion emit more than a vehicle moving quickly in a non-congested area?

(3) Line 204: What was the typical roadway length? For roadways with a length greater than 2 km, how was the average speed calculated, especially if some parts have slow traffic and others have fast traffic?

(4) Line 171: Is reference 21 the original source of the data for Table 1?

(5) Line 204: How was the average speed for small intervals (e.g., 15 minutes) calculated from the raw online data?

(6) Figures 6-9: VOC and CO emissions appear to be quite constant in these figures. Are there better ways to present these results for CO and VOC emissions?

(7) Line 226: What/where is the S3 table?

6. PLOS authors have the option to publish the peer review history of their article (what does this mean?). If published, this will include your full peer review and any attached files.

Reviewer #1: No

Reviewer #2: No

Reviewer #3: No

---

## [Author Response · Author response to Decision Letter 0]

15 Nov 2023

Reviewer #1: 

Question 1. The map of China in Figure 4 is incomplete, and the South China Sea Islands are omitted.

Author response: We have corrected Figure 4 to a complete map of China and re-uploaded the corrected Figure 4 upon submission.

Question 2. What does “EF” in the second paragraph of the introduction mean?

Author response: "EF" means emission factor, which has been corrected.

Question 3. The organization of the paper is somewhat chaotic. The last second paragraph of the introduction points out that Section 2 describes the data and data processing. But the following headline is "Methodology".

Author response: We have revised the last paragraph of the introduction. The revised content is Section 2 describes the data collection, data processing, and emission estimation methods for this study.

Question 4. The authors should go through the paper for typos and syntax errors. There are many grammatical errors and expressing inconsistencies in the paper. For example: The symbols “-” and “~” in the sentence “0.220-0.253g/veh, 0.005-0.01g/veh, 0.018324 ~0.047g/veh, 133.78~275.21g/veh” are inconsistent.

Author response: We have revised the grammatical errors and expressed inconsistencies in the paper

Question 5. What are the innovations of the paper? What are the contributions to this research field?

Author response: 

(1) The innovation: This paper proposes a method for estimating traffic emissions using big data, using the proposed traffic emission intensity index to quantify traffic emissions from the road level and study regional spatiotemporal distribution patterns.

(2) The contributions: This paper uses Baidu Map online data and OSM data can reduce the cost of data collection and processing, and provides new ideas for urban traffic emission research.

Question 6. The first paragraph of the introduction is too simple, it is recommended to rewrite it.

Author response: We have rewritten the first paragraph of the introduction.

Reviewer #2: 

This paper introduces a methodology for using big data sources to estimate emission intensity on roadways. The proposed methodology can replace the expensive methods of evaluating roadway emissions. Based on my review of this paper, here are my comments:

Question 1. Line 181 In equation 3, why is emission intensity directly related to length? What does this relationship mean? What is the unit of "Hp g/veh-km"? If the average speed is the same, would a road with a 2 km link have double the emission intensity of a road with a 1 km link?

Author response: 

(1) In equation 3, emission intensity quantifies the emissions of vehicles passing through the road section at a certain operating speed from the perspective of the road section. This indicator is the product of the vehicle emission factor based on vehicle speed and the distance traveled. Therefore, emission intensity is directly related to the length of distance traveled.

(2) The unit of traffic emission intensity is g/veh, this part of the content has been modified.

(3) This means that distance length is proportional to emission intensity at the same average speed. When the average speed is the same, the emission intensity of a 2-kilometer-long road is twice that of a 1-kilometer-long road.

Question 2. Line 87: Since traditional GPS-based emission analysis requires heavy data processing and specialized training, how easy is it to use the proposed method? If decision-makers intend to use the proposed methodology, how many days would it take to complete the analysis?

Author response: 

(1) The method in this article can obtain full coverage speed data of each road in real-time through Baidu online map API. This method reduces the dependence on traffic volume data and can complete processing and analysis in a short time through convenient analysis methods, which is conducive to the rapid development of research.

(2) For decision-makers, the time cost of data analysis is determined by the data cycle. For example, an analysis of road emission intensity over a week would require data acquisition for a week.

Question 3. Line 38: In the abstract, mention the actual number or percentage of segments on Second and Third Street instead of "some segments."

Author response: We have modified the content of the abstract based on review comments

Question 4. Line 171: Please cite the original data source for Table 1.

Author response: We have correctly cited the original data source for Table 1.

Question 5. Line 208: What percentage of the traffic data was original, and what percentage of data was missing that required linear interpolation?

Author response: The original percentage of the traffic data was 99%-100%. Only one or a few data are missing due to network fluctuations, and the percentage of data requiring linear interpolation does not exceed 1%.

Question 6. Line 211: Provide further details on how Equation 4 was used in the case study. Was the average speed for every 15 minutes used?

Author response: Equation 4 is used to calculate road traffic emission intensity at the hourly level. We calculated the value of hourly traffic emission intensity for 153 roads over 24 hours using Equation 4 in the case study and have modified it in the article (lines 288-289 of file ‘Revised Manuscript with Track Changes’). The average speed for every 15 minutes was used for hourly traffic emission intensity calculation.

Question 7. Figures 6-9: Why is CO emission almost constant regardless of the time of day? Is CO emission not affected by slowdowns during peak hours?

Author response: 

(1) In this study, the proportion of CO and VOC in vehicle exhaust is very small, so the changes in emission intensity are not obvious.

(2) During peak hours, carbon monoxide emissions are affected by deceleration. When the speed is low, insufficient fuel combustion increases CO emissions.

Reviewer #3: 

This paper identifies a gap in the existing literature related to using big data regarding traffic to evaluate traffic emissions. It aims to fill this gap by utilizing Baidu traffic data and the COPERT model to estimate traffic emissions.

Question 1. It's not clear what "emission intensity" means. Are we trying to identify roadways where emissions from a single vehicle are high? This reasoning seems to support not using traffic volume as a factor.

Author response: 

(1) Emission intensity refers to the total traffic emissions of vehicles passing through the road at the average speed of the road from the perspective of the road segment.

(2) We are trying to use the emissions of vehicles passing that road at the average road speed to reflect the traffic emissions on each road in the urban road network.

(3) This study attempts to omit the collection of traffic volume data and only estimate emissions by obtaining the average speed of vehicles on urban roads online, reducing data collection costs.

Question 2. Is emission intensity (emissions from a single car) high for congested roadways? Why does a vehicle stuck in congestion emit more than a vehicle moving quickly in a non-congested area?

Author response: 

(1) The emission intensity (single-vehicle emissions) of congested roads needs to be determined based on the average driving speed of vehicles and the length of the congested road section.

(2) The reason is that vehicles in congestion are traveling at low speeds and insufficient fuel combustion leads to increased exhaust emissions.

Question 3. Line 204: What was the typical roadway length? For roadways with a length greater than 2 km, how was the average speed calculated, especially if some parts have slow traffic and others have fast traffic?

Author response: 

(1) Due to the differences in the length of urban roads, typical road lengths are between 1km and 2km.

(2) For urban roads with different lengths and traffic congestion conditions, the average speed is calculated in the same way, that is, using Baidu Maps Online API to crawl the road length and vehicle travel time through the road, and then obtain the average speed.

Question 4. Line 171: Is reference 21 the original source of the data for Table 1?

Author response: The original source of the data for Table 1 is reference 29, which we have corrected.

Question 5. Line 204: How was the average speed for small intervals (e.g., 15 minutes) calculated from the raw online data?

Author response: Taking the 15-minute interval as an example, we used the compiled program to obtain the road length and travel time every 1 minute through Baidu Online Map API, and obtained the speed by calculating the quotient of road length and travel time. The average of all speeds over a 15-minute period was considered the average speed.

Question 6. Figures 6-9: VOC and CO emissions appear to be quite constant in these figures. Are there better ways to present these results for CO and VOC emissions?

Author response: In this study, the proportion of CO and VOC in vehicle exhaust is very small, so the changes in emission intensity are not obvious. The histogram is to more intuitively present the proportion of emission intensity of different types of pollutant gases in the overall emission intensity.

Question 7. Line 226: What/where is the S3 table?

Author response: The S3 table is in the S3_Table file of Supporting Information

---

## [Decision Letter · Decision Letter 1]

28 Nov 2023

Data Mining and Spatio-temporal Characteristics of Urban Road Traffic Emissions: A Case Study in Shijiazhuang, China

PONE-D-23-14153R1

Dear Dr. Wu,

We’re pleased to inform you that your manuscript has been judged scientifically suitable for publication and will be formally accepted for publication once it meets all outstanding technical requirements.

Kind regards,

Xiangjie Kong

Academic Editor

PLOS ONE

Additional Editor Comments (optional):

Thank the authors for the efforts to improve the work. The current version successfully satisfied all reviewers.

Reviewers' comments:

Reviewer's Responses to Questions

**Comments to the Author**

1. If the authors have adequately addressed your comments raised in a previous round of review and you feel that this manuscript is now acceptable for publication, you may indicate that here to bypass the “Comments to the Author” section, enter your conflict of interest statement in the “Confidential to Editor” section, and submit your "Accept" recommendation.

Reviewer #1: All comments have been addressed

Reviewer #2: All comments have been addressed

Reviewer #3: All comments have been addressed

2. Is the manuscript technically sound, and do the data support the conclusions?

Reviewer #1: Yes

Reviewer #2: Yes

Reviewer #3: Yes

3. Has the statistical analysis been performed appropriately and rigorously? 

Reviewer #1: Yes

Reviewer #2: Yes

Reviewer #3: Yes

4. Have the authors made all data underlying the findings in their manuscript fully available?

Reviewer #1: Yes

Reviewer #2: Yes

Reviewer #3: Yes

5. Is the manuscript presented in an intelligible fashion and written in standard English?

Reviewer #1: Yes

Reviewer #2: Yes

Reviewer #3: Yes

6. Review Comments to the Author

Reviewer #1: (No Response)

Reviewer #2: This paper proposed an approach that can be used to study and nalyze the spatio-temporal emission pattern of a region at a roadway section level by using Baidu's online traffic data and COPERT model.

The road traffic status is obtained through Baidu online data, and the line reference model of GIS technology is used to match the traffic data and road data to complete the preparation and processing of basic data. Combining the COPERT emission model to calculate the emissions of each road in the urban road network, a traffic emission intensity index is proposed to quantify the road emissions, and the traffic emission intensity of the urban road network is analyzed from the perspective of time and space.

The research results are of great significance to the evaluation of urban road traffic emissions and the government's formulation of carbon neutral measures and also provide a novel idea for traffic emissions research.

Reviewer #3: This article studies the urban traffic emissions problem in the transportation field, proposes a method to evaluate road-level emissions by combining map online data and the COPERT emission model, and uses actual data for verification analysis.

The proposed method can avoid the problem of strong reliance on traffic volume data that exists in most emission studies, helping to reduce data acquisition and analysis costs.

This paper proposes a traffic emission intensity index to quantify road emissions, and on this basis, analyzes the temporal and spatial distribution characteristics of urban road traffic emission intensity.

The structure of the article is neat and the content is relatively clear. The results of the article will help provide new methods for urban traffic emission research and play a positive role in promoting the development of urban green and low-carbon transportation.

7. PLOS authors have the option to publish the peer review history of their article (what does this mean?). If published, this will include your full peer review and any attached files.

Reviewer #1: **Yes: **Zhenzhen Yang

Reviewer #2: No

Reviewer #3: No

---

## [Editor Report · Acceptance letter]

2 Dec 2023

PONE-D-23-14153R1 

Data Mining and Spatio-temporal Characteristics of Urban Road Traffic Emissions: A Case Study in Shijiazhuang, China 

Dear Dr. Wu:

I'm pleased to inform you that your manuscript has been deemed suitable for publication in PLOS ONE. Congratulations! Your manuscript is now with our production department. 

Kind regards, 

on behalf of

Dr. Xiangjie Kong 

Academic Editor

PLOS ONE